# Preparation of Nano Silicon Carbide Modified UV Paint and Its Application Performance on Wood Flooring Surface

**DOI:** 10.3390/polym15234584

**Published:** 2023-11-30

**Authors:** Kankan Zhou, Manping Xu, Wangjun Wu, Jin Wang

**Affiliations:** 1Zhejiang Academy of Forestry, Key Laboratory of Bamboo Research of Zhejiang Province, Hangzhou 310023, China; kankan022@163.com (K.Z.); xump666@163.com (M.X.); wwj18351869585@163.com (W.W.); 2College of Chemistry and Materials Engineering, Zhejiang A&F University, Hangzhou 311300, China

**Keywords:** super-abrasion-resistant, coating, paint, UV lacquer, wood flooring, nano silicon carbide

## Abstract

This study aims to tackle the drawback of non-abrasion-resistance of wood flooring with paint finish. A new method for preparing wood flooring with super-abrasion-resistant coatings by adding nano silicon carbide (SiC) particles to the paint was developed. As indicated by the results, the best mass fraction of nano SiC powder added in ultraviolet (UV) paint is 2.0%, the suspension liquid is stable when the mass concentration of sodium hexametaphosphate added is 2.5%, and it is better for the site humidity to remain below 75% when the nano SiC paint coating is applied. During the preparation of wood flooring with super-abrasion-resistant coating finish, the dosage of finish applied each time should not exceed 30 g/m^2^. During the sanding process, the sanding speed needs to be increased by about 2 m/s compared with that for the ordinary UV nano SiC primer in production. The test results of the performance of finished products indicate that the prepared wood flooring has better film abrasion resistance, adhesion of paint film, and film hardness. Meanwhile, because the paint film is durable and weather resistant, the service life of flooring is effectively extended, avoiding a significant waste of the product’s use value and broadening the product’s application range.

## 1. Introduction

Wood flooring with paint coating mainly include solid wood flooring, engineered wood flooring, glued laminated bamboo flooring, and other varieties. These kinds of flooring look bright and glossy, while maintaining the advantages of wood such as the natural wood patterns and a comfortable feeling on feet. However, the paint flooring come with some downsides such as poor abrasion-resistance and scratch-resistance on the surface. Especially when compared with impregnated paper laminated wood flooring, the paint wood flooring has lower abrasion-resistance on the surface.

The major drawback of non abrasion-resistance of wood flooring coatings is reflected in the fact that after being used in households for a few years, the surface coating of solid wood flooring tends to be worn out and starts to fade, eventually revealing the wood beneath it and causing the flooring to lose its original beauty. The flooring with a worn-off protective layer not only lose their luster, but also are prone to decay, significantly wasting the use value of wood flooring.

The development of super-abrasion-resistant coatings is crucial for the development of the wood flooring industry. At present, the surfaces of many types of solid wood flooring such as solid wood flooring and engineered wood flooring are decorated with ultraviolet (UV) cured paint. The paint surface is full and smooth, bringing out satisfactory results in appearance. As people’s living standard improves, engineered wood flooring are increasingly expected to maintain their advantages without compromising the outstanding surface performance such as abrasion resistance, toughness, and scratch resistance exhibited by laminate wood flooring which imposes higher requirements on the quality of the paint film applied on solid wood flooring. In recent years, the research and application of nanomaterials have become a hot topic in various industries. With ability to improve coatings’ adhesion, abrasion resistance, impact resistance, flexibility, corrosion resistance, and radiation resistance [1], nano coatings have vast development prospects in the coating industry [2,3,4]. Consumers’ confidence in purchasing wood flooring can be boosted if the abrasion resistance of the protective surface layer of solid wood flooring can be improved through the development of the super-abrasion-resistant coatings of wood flooring. Such a development is also an important condition for the development of the wood flooring industry.

Current research of super-abrasion-resistant coatings largely focuses on application of abrasion-resistant layers that contain Al_2_O_3_ on the surface coating [5,6], application of super-hard resin coating finish [7], improvement of paints to allow paint particles to penetrate into the fiber pores of wood [8], use modification of resin coatings using nanoparticles [9,10,11], and enhancement of the abrasion resistance of wood flooring by using fast-drying high-performance waterborne UV wood coatings and through processes such as combining microwave, infrared, and hot air in drying [12].

SiC materials have a series of advantages such as excellent mechanical properties, high thermal conductivity, high field breakdown strength, and excellent physical and chemical stability [13]. Liao et al. implanted high-energy carbon ions into boron doped silicon wafers through ion implantation method. The silicon wafers were annealed at 950 °C in nitrogen gas, and SiC nanoparticles were generated on the surface of the wafers. This material was then etched into a porous structure to prepare blue light emitting devices [14]. Ortona et al. prepared SiC_f_/SiC composites by combining CVI and PIP processes to leverage the advantages of different processes and reduce costs [15]. This article presents a novel method for preparing wood flooring with super-abrasion-resistant coatings, which is a new study different from the aforementioned literature. In this study, the nanomaterials were connected and bonded with macro-molecular chains in paint with strong surface force and bonding force by leveraging their special surface effect. Thus, it is necessary to keep the flexibility of the macro-molecular chains while dramatically enhancing the adsorbing force between the paint and the wood surface. The small size effect of nano SiC particles ensures that the paint’s film-forming molecules are closely connected, making the surface of the paint film more dense, solid, and delicate, thereby improving the abrasion resistance of the paint. Additionally, due to the high level of hardness of SiC, abrasion resistance can still be improved even if a small quantity of nanoparticles agglomerate. Therefore, nano SiC paint offers a quality foundation for developing wood flooring.

## 2. Materials and Methods

### 2.1. Materials and Instruments

The test sample is a multi-layer structural composite flooring with a specification of 910 mm × 125 mm × 15 mm (length × width × thickness), a surface panel of oak (thickness 0.6 mm), an intermediate substrate layer of eucalyptus plywood, and a bottom plate of birch veneer, which was provided by Hangzhou Mingcheng Wood Industry Co., Ltd. (Hangzhou, China). The moisture content of the flooring is 9.5%. Ethylene glycol was provided by Shanghai McLean Biochemical Technology Co., Ltd. (Shanghai, China). The UV paint was provided by Hunan Changsha Xinkai Chemical Co., Ltd. (Changsha, China). The UV paint is mainly composed of polyurethane acrylate (C_4_H_6_NO_2_)_n_(C_3_H_4_O_2_)_m_, with a content of about 55%,was provided by Hunan Changsha Xinkai Chemical Co., Ltd. (Changsha, China). The other components include acrylic monomers (about 25%, tripropylene glycol diacrylate, trimethylolpropane triacrylate, 1,6-ethylene glycol diacrylate, hydroxyethyl methacrylate, Hunan Changsha Xinkai Chemical Co., Ltd., Changsha, China), talc powder (about 8%, Hangzhou Huipu Chemical Co., Ltd., Hangzhou, China), aromatic ketone initiators (about 5%, 2-hydroxy-2-methyl-1-phenyl-1-acetone, 1-hydroxycyclohexylphenyl ketone; Changzhou Qiangli Electronic New Materials Co., Ltd., Changzhou, China), polysiloxane defoamers (about 2%, (R_2_SiO)_x_, industrial grade, Guangdong Zhonglianbang Fine Chemical Co., Ltd., Dongguan, China), anti precipitation agents (about 3%, polyhydroxyamide polymer, industrial grade, Dongguan Siye Plastic Co., Ltd., Dongguan, China), leveling agents (about 2%, Polydimethylsiloxane, industrial grade, (C_2_H_6_OSi)_n_, Shanghai Deyude Trading Co., Ltd., Shanghai, China), etc. The viscosity of the UV paint used for coating was 1000–2000 mpa·s. The nano-SiC was purchased from Shanghai Chaowei Nanotechnology Co., Ltd. (Shanghai, China). The average particle size of nano silicon carbide is 40 nm, with a purity greater than 99.9% and a specific surface area of 39.8 m^2^/g. The putty was provided by Sankeshu Coatings Co., Ltd. (Putian, China). The viscosity of the putty was 10,000–13,000 mpa·s. The fully automatic homogenization system was from RayKol (California, USA). The universal mechanical testing machine was from Shimadzu (Kyoto, Japan). The surface roughness measuring equipment was from Mitutoyo (Kagawa, Japan). The paint film grinding instrument was from Shanghai Rongjida Instrument Technology Co., Ltd. (Shanghai, China). 

### 2.2. Preparation of Nano-SiC Paint

In the test, ethylene glycol was used as the solvent. First, nano-SiC was stirred and dispersed using an automatic homogenization system. Then, the nano-powder slurry with uniform dispersion and small particle size was obtained by grinding. Finally, the slurry was added to the UV paint based on the calculated proportion. 

The specific steps are as follows (Figure 1):

Stirring and dispersion: Some solvents and dispersion resins were stirred evenly, and then added with dispersants, wetting agents, etc. After becoming completely dissolved and uniform, the nano-powder was gradually joined and stirred in the automatic homogenization system for 1 h. Grinding and dispersion: The resulting composite powder slurry was added with a certain amount of defoamers and wetting agents, and ground for 1 h in a stirred ball mill through wet grinding to obtain nano-SiC powder slurry. Preparation of paint: The resulting nano-SiC powder suspension was added to UV paint in a calculated proportion, stirred slowly. Next, dispersants, defoamers, and other additives were added and mixed thoroughly to obtain the finished product. 

The amount of nanomaterials added during the preparation of nano-SiC paint is one of the factors to be considered carefully, because insufficient addition can not produce enough modification efficiency, and excessive addition not only increases the production cost of paint but also exacerbates the nanomaterials tendency to aggregate, affecting their dispersity and stability in paint and weakening the performance of modified paint. It has been reported that the amount of nanomaterials added to coatings should be kept in a range from 1% to 3% of the total coating mass [16,17,18]. In the test, the amount of SiC added was 1.5%, 2.0%, and 2.5% of the total coating mass. Trial production of super-abrasion-resistant flooring using the same batch of engineered wood nano flooring was conducted to study the effects of different amounts of nano-SiC powder added on the performance of the modified paint. Each level was repeated three times.

### 2.3. Preparation of Super-Abrasion-Resistant Wood Flooring with Nano-SiC Paint

Wood material sanding was first performed. This process aims to set the thickness of the flooring slab and to polish the flooring surface. The effect of material sanding directly affects the quality of the flooring and the amount of paint used per unit area. The nano-SiC special primer was sprayed at a dosage of 8–12 g/m^2^ and semi-cured by UV. Its function is to improve the adhesion between the paint and the wood flooring, as well as to prevent cracking. Putty application was carried out. The function of putty is to fill in the wood grain capillaries, grooves, etc., on the surface layer of flooring, creating a flat flooring surface. But for tree species with thick grains such as red oak and oak, putty application is often necessary to conduct for multiple times to ensure that the capillary of the surface layer on the flooring is 100% filled. Transparent putty was used at a dosage of 20–30 g/m^2^ and cured by UV. Sanding was performed with belt 320 # or 280 #, which was used to sand the uneven surface to achieve the effect of paint filling. The adhesion between paints and that between paint films were improved. The nano-SiC hardening primer was sprayed at a dosage of 15–20 g/m^2^ and semi-cured by UV. Sanding primer was sprayed at a dosage of 10–15 g/m^2^ onto the surface of nano-SiC abrasion-resistant primer first since the nano-SiC abrasion-resistant primer has high strength, followed by UV curing. Sanding was implemented. Sanding belt 320 # was used for sanding, and its operating speed was appropriately increased to enhance the sanding effect since the nano-SiC abrasion-resistant primer is hard. After another cycle of “putty application/sanding/nano-SiC abrasion-resistant primer spraying/sanding primer spraying/sanding” was conducted to enhance the hardness and abrasion resistance of the surface of the flooring. A specialized primer for the process of flow coating-roller coating was sprayed prior to flow coating—roller coating of nano-SiC paint. Flow coating of nano-SiC paint was carried at a dosage of 110 g/m^2^, followed by UV curing. This process can make the flooring appear visually plump. The nano-SiC abrasion-resistant primer was applied once after sanding, Then, sanding primer was sprayed, followed by sanding. Thereafter, the finish coat was applied. Polishing was conducted on the dust removal machine with the two brush rollers changed to two polishing wheels. Such a change can make the machine meet the requirements of dust removal and achieve the polishing effect. Scratch-resistant nano-SiC finish coat was applied at a small amount but high frequency (it was applied three times, with a dosage of 8 g/m^2^ each time) to achieve a delicate and plump effect.

The nano-SiC paint is based on UV paint, and its UV curing nature requires that the site environment be clean and tidy. Generally, the ambient temperature should not be too low. The requirements for humidity need to be further studied as there are distinctions between different paints. Paint films (with a thickness of 2 mm) were prepared on glass panels at relative humidity levels of 45%, 55%, 65%, 75%, and 85%, respectively. Each level was repeated three times.

The effect of the coating dosage of nano-SiC paint on the performance of paint film was studied by comparing the film abrasion resistance and adhesion of paint films of flooring with paint coated at different dosages. In the production of engineered wood flooring, the maximum coating usage of UV paint is usually controlled below 200 g/m^2^, based on which the corresponding experimental level was determined. Using the three-step method of sanding, roll-painting &UV curing, the plain board of engineered wood flooring that use sesame bean panel (2 mm) was first painted with nano-SiC primer once and then with nano-SiC finish coat once. Following each roller coating, curing was conducted.

The abrasion resistance of paint film reflects the paint quality of engineered wood flooring. Generally speaking, flooring with a lower abrasion value have better abrasion resistance. However, the abrasion value of paint film can only reflect a relative quantity and cannot reflect the final product life. The film abrasion resistance can better reflect the flooring durability and is currently one of the indicators most valued by flooring manufacturers and consumers. The adhesion of paint film refers to the ability of paint film to bond with the surface of the painted object. Poor adhesion can lead to quality risks such as blistering on surface paint, cracking, and even peeling during use, affecting the flooring service life. The film hardness is an indicator of the mechanical strength of paint film. Low film hardness means that flooring have poor impact resistance and are prone to scratches and abrasion.

Sanding is the foundation to ensure the quality of the paint surface, and the sanding effect is directly associated with the smoothness, glossiness, and decoration effect of the surface of the paint film. As soon as each layer of primer dries, it should be sanded in a timely manner to reduce surface roughness and to create a clean, smooth, and flat coating surface, thereby effectively improving the paint film’s adhesion and the paint appearance.

Surface roughness can quantitatively characterize the quality of sanding, and the optimal sanding process conditions for nano-SiC paint can be studied by measuring the roughness of the surface of primed boards under different sanding conditions. In this test, a surface roughness measuring instrument was used for measurement. The roughness at six random points on each specimen was measured and the average was taken. The sampling length of the instrument was 2.5 mm, and the measuring length was 10 mm, which was four times that of the sampling length. While measuring the surface roughness, the movement direction of the sensor remained perpendicular to the sanding direction of the specimen. Sand belt 400 # (Represented by the step 11 in Figure 2) was selected to compare the impact of the sanding belt’s operating speed on the surface roughness of the flooring’s nano-SiC primer.

Due to the addition of hard SiC, the nano-SiC paint has its own characteristics as well as higher requirements when it comes to sanding. The sanding effect can be improved by adjusting the operating speed of the sanding belt.

### 2.4. Method for Testing the Performance of SiC Paint Film

The paint film’s three parameters, namely tensile strength, elongation at break, and tear strength, were tested according to GB/T 528-2009 and GB/T 529-2008 [19,20]. The abrasion resistance of the paint film of the super-abrasion-resistant flooring from trial preparation was tested based on the test method specified in Section 4.46 of GB/T 17657-2022 [21]. Abrasion value was tested according to Section 4.47 of GB/T 17657-2022. The performance of the surface resistance to cycled cold and heat was tested following the test method specified in 4.41 of GB/T 17657-2022, but the number of cycles was changed from two to ten. The adhesion was tested using the test method specified in GB/T 4893.4-2013 [22]. The humidity was the relative air humidity during application and curing. Film hardness was tested using the test method specified in GB/T 6739-2022 [23]. Finally, the surface appearance of the paint film was tested by visual inspection. Each level was repeated three times.

### 2.5. Characterization

The surface morphology was characterized by scanning electron microscopy (SEM, Quanta 200, FEI, Hillsboro, USA). Synchronous thermal analysis (TG-DSC, STA409PC, Selbu, Germany) was employed to determine the variation in sample quality and the relevant information on the absorption and release of heat. Then, 15 mg of sample powder was heated in a nitrogen protection atmosphere at a rate of 10 °C/min, ranging from 30 °C to 650 °C. Three replicates were performed in each group.

## 3. Results and Discussion

### 3.1. Impact of Ratios of Nano-SiC Powder Added in UV Paint on Performance of Paint Film

The test results of paint film performance are shown in Table 1. It can be seen that after adding nano-SiC powder, the performance of the coating’s paint film was significantly improved, especially in terms of film abrasion resistance and abrasion value. According to comparison results of the test indicators of each sample, it can be determined that the optimal mass fraction of nano-SiC powder added is 2.0%.

### 3.2. Impact of Dispersant Dosage on Paint Stability

From Figure 3, it can be seen that when there was a small dosage of dispersant, the dispersion effect improved notably with the increase of dispersant dosage. When there was a large dosage of dispersant and the dosage continued to increase, the dispersion effect actually deteriorated, and an extreme point appeared in Figure 3.

In terms of the dispersant dosage, a mass concentration of sodium organophosphate of 2.5% allowed the suspension liquid having a smaller particle size and becoming more stable when pH was 6 and the mass fraction of SiC was 2% [10,24].

### 3.3. Impact of Humidity on Paint Film

From Table 2 and Figure 4, it can be concluded that nano-SiC paint coating does not have high requirements on temperature and the relative air humidity during application and curing. However, under high humidity, the paint film tended to whiten, affecting the decoration effect, and occasionally there were drawbacks such as blistering and pinholes. During the application of nano-SiC paint, it is necessary to control humidity to a certain extent, and it is better to keep the site humidity below 75%.

### 3.4. Impact of Coating Dosage on Performance of Paint Film

From Figure 5, it can be seen that the abrasion resistance of the paint film improved as the dosage of nano-SiC paint applied increased and that the two were linearly correlated. It can also be seen that when the nano-SiC paint was fully cured, there was no difference in abrasion resistance between its surface and its interior. The larger the coating dosage was, the better the abrasion resistance would be.

From Table 3, it can be seen that the film hardness is not closely related to the coating dosage, but largely determined by the characteristics of the nano-SiC paint itself. The adhesion of paint film rapidly decreased as the coating dosage of nano-SiC paint increased. During UV curing, the paint that was far from the substrate surface was subjected to more intense UV radiation, resulting in a higher rate of shrinkage than the paint near the substrate surface. The larger the coating dosage, the more notable the curing shrinkage, which affected the paint’s penetration into the wood fiber pores, thereby weakening the adhesion performance of paint film.

Excessive application of nano-SiC paint increases production costs and adversely affects the paint film’s overall performance. Application at a small dosage for multiple times and sanding can make the paint fully penetrate into the wood, effectively enhancing the bonding force between the paint and the flooring. When paint was applied near the substrate, the dosage in each application should not exceed 30 g/m^2^. When the total coating dosages are almost the same, the only way for flooring to maintain a high grade of adhesion of paint film is to enhance the abrasion resistance of the paint film by improving the process and the quality of paint.

### 3.5. Impact of Sanding Process on Performance of Paint Film

The sanding step is to grind and smooth the finishing of the primer grinding and smooth finishing of the primer, which is beneficial to enhance the adhesion of the film and reduce the amount of paint. From Figure 6, it is found that the impact of the number of revolutions of the sanding belt on surface roughness. Compared with that for ordinary UV primer, the operating speed of the sanding belt needs to be increased by about 2 m/s in production in the case of nano-SiC primer. An increased operating speed of the sanding belt enhances the sanding grinding force. Under the same operating speed of the sanding belt, the roughness of nano-SiC primer is higher than that of ordinary UV primer, because the sanding belt needs to run faster in production as nano-SiC primer has a higher level of hardness.

### 3.6. Quality Performance of Wood Flooring

As the particle size becomes smaller, the ratio of the number of nanoparticle surface atoms to the total number of atoms rises sharply. Due to the lack of adjacent coordinating atoms on the surface, a surface energy different from that in conventional particles exists between nanoparticles. In this case, the valence bonds are unsaturated and have a tendency to bond with external atoms, resulting in high activity. Because of such a surface energy, the nanoparticles have a unique property that enables them to improve the abrasion resistance of paint. However, this characteristic also makes the nanoparticles susceptible to agglomeration, thereby forming secondary particles with enlarged sizes and causing particles to lose the characteristics possessed by the nanoparticles. Therefore, the dispersion of nano-SiC plays a key role in improving the performance of paint.

Excellent performance has been witnessed in the quality checking of the trial-produced engineered wood flooring product with SiC paint. The test results of the performance of the flooring products’ paint film are shown in Table 4.

### 3.7. Microscopic Morphology Analysis

The SEM analysis was performed on the surfaces of wood flooring substrates, conventional primer coated flooring, conventional paint coated flooring, and flooring modified with nano silicon carbide paint. The results are shown in Figure 7. It can be seen from Figure 7a,b that the substrate material of the wooden flooring is unpainted and has a relatively smooth surface. The typical wood cross-sectional morphology can be seen, and the surface coating of the panel is smooth and uniform. Wong et al. [25] reported that the elastic modulus E of SiC nanowires with a diameter of 20–30 nm can reach 600 GPa, and the maximum bending strength can reach 53.4 GPa, which is approximately one tenth of the elastic modulus E. From Figure 7c,d, it can be seen that after the material wood is coated with paint primer and putty, the surface is sanded, and there are many small fragments present on its surface. For smaller SiC nanoclusters, there is almost no difference in the crystal structure and electronic structure of different polytypes of SiC [26]. After further finishing treatments, as shown in Figure 7e,f, the surface has been filled, further reducing the roughness and improving the overall effect, but still showing some fragments. The surface is coated with silicon carbide modified paint, as shown in Figure 7g,h. There are relatively uniform nanoscale particles on its surface, which can be seen in the range of 1–5 μm through magnification images. The nanoparticles can better fill the pores of the veneer completely, significantly improving the surface wear resistance of wood veneer flooring.

### 3.8. Thermal Stability Analysis

The thermo-gravimetric analysis was conducted on four materials: wood flooring substrate, conventional primer coated flooring, conventional paint coated flooring, and modified paint flooring with the addition of nano silicon carbide. The results are shown in Figure 8. From Figure 8a, it can be seen that all four samples have an appropriate weight reduction around 100 °C, with the wood material showing the most significant reduction. This is because the moisture content of the wood material is higher than that of other paint coated wood, resulting in a greater loss of moisture in the early stage. The three types of painted wood show significant quality loss in the range of 250 °C to 450 °C, and the wood material exhibits significant decomposition in the range of 240 °C to 360 °C, indicating that painting can improve the stability of wood and delay quality loss. After the temperature rises to 650 °C, the quality loss of the sample A, B, C and D are 23.4%, 15.3%, 9.4%, and 25.7%, respectively. The quality loss of the wear-resistant paint flooring treated with silicon carbide modification has the highest residue rate. Silicon carbide paint contains high inorganic components, resulting in a higher final residue rate.

As shown in Figure 8b, the overall trend of the heat release pattern of the four samples is relatively consistent. Overall, the untreated initial wood has the highest heat release, and its heat release has decreased after painting. The statistic heat-resistant index (Ts) of samples A, B, C and D were calculated with the relation [27]: Ts = 0.49[T5% + 0.6(T30% − T5%)]. The results of the above four samples are 107, 142.5, 150.3, and 156.6, respectively. Compared to the four types of wood samples, the sample D of paint flooring modified with silicon carbide nano coating has higher thermal stability and the lowest heat release rate. Therefore, the wear-resistant flooring modified by silicon carbide not only has stronger wear resistance on its surface, but also improves its thermal stability.

## 4. Conclusions

The results show that the properties of the coating film are obviously improved after adding nano SiC powder in the preparation of nano SiC paint. The optimum mass fraction is determined to be 2% by comparing several groups of different nano SiC powder addition ratios. In terms of the amount of dispersant, when pH = 6, the optimum mass fraction of sodium hexametaphosphate was 2.5%. When nano SiC paint is applied, the relative humidity of the construction environment should not exceed 75%. When coating close to the substrate, the coating amount should not exceed 30 g/m^2^ each time. Compared with ordinary UV primer, the running speed of nano SiC primer needs to be increased by about 2 m/s when sanding.

This kind of wood flooring has better film abrasion resistance, adhesion of paint film, and film hardness. This paint film is more durable and weather resistant and can better protect engineered wood flooring. This study has resolved the following problems encountered by the currently available paint films including poor durability and inability to maintain good film hardness and adhesion at the same time. This paint film has effectively extended the service life of flooring and avoided enormously wasting the use value of wood flooring. Compared with the other ordinary UV paint, this nano-SiC paint enables engineered wood flooring to display a more prominent texture. The unique characteristics such as super abrasion resistance and scratch resistance have also extended the flooring service life as a whole, while expanding their application range.

## Figures and Tables

**Figure 1 polymers-15-04584-f001:**
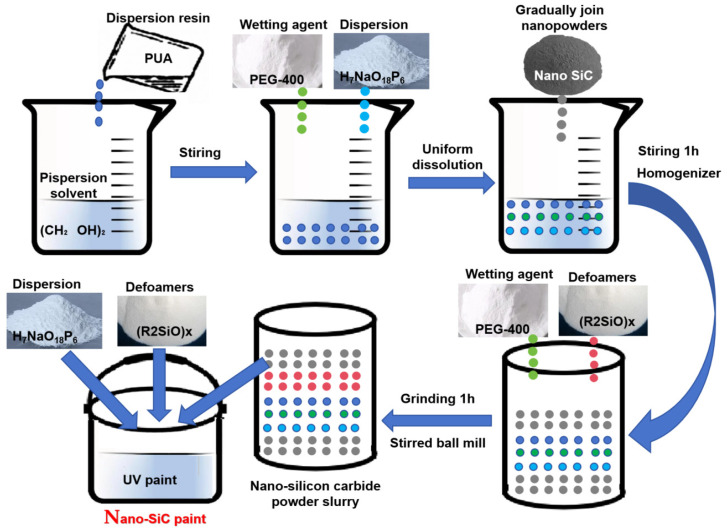
Schematic illustration of the synthesis procedure of the nano-SiC paint.

**Figure 2 polymers-15-04584-f002:**
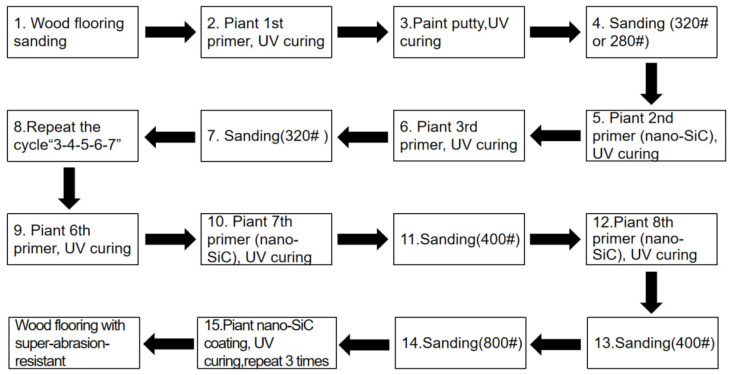
Schematic illustration of the process of the wood flooring coated with the nano-SiC paint.

**Figure 3 polymers-15-04584-f003:**
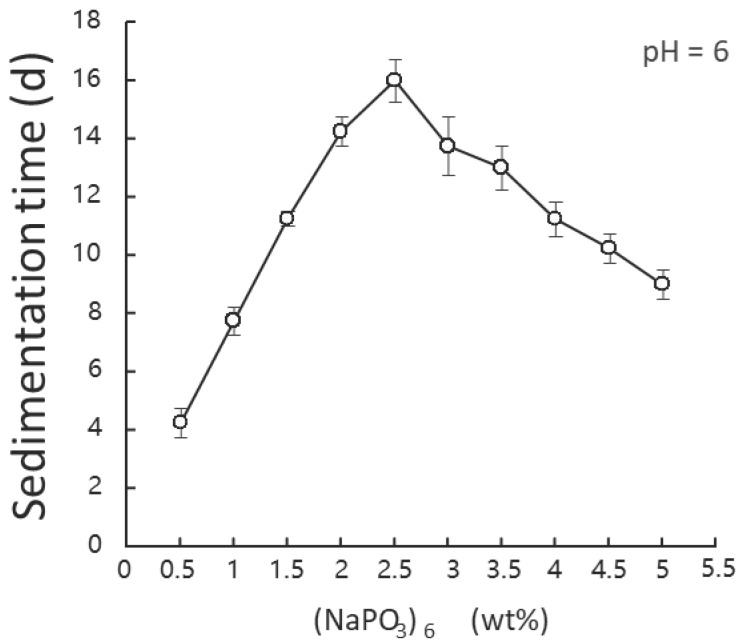
Impact of dispersant dosage on sedimentation speed.

**Figure 4 polymers-15-04584-f004:**
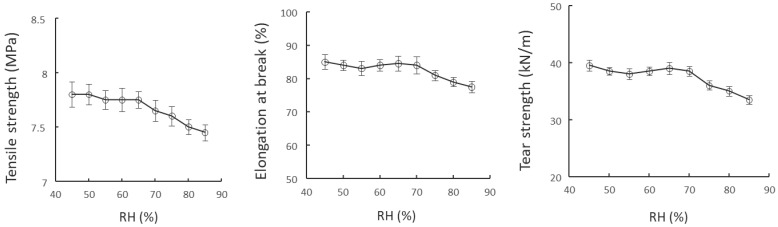
Impacts of different humidity levels on the mechanical properties of nano-SiC paint film.

**Figure 5 polymers-15-04584-f005:**
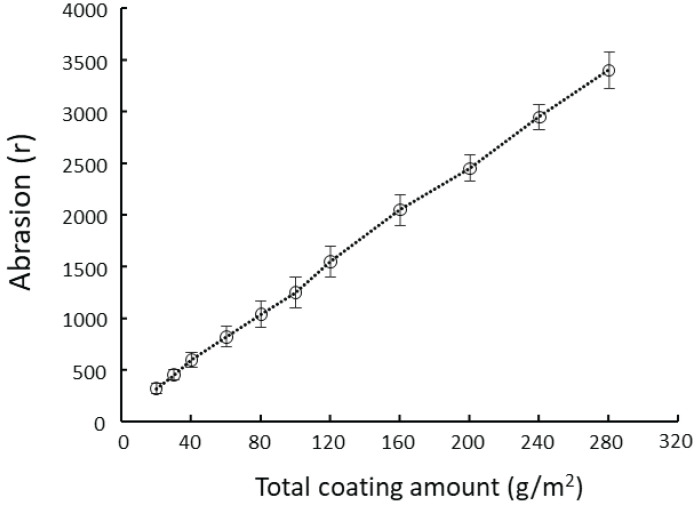
Impact of coating dosage on film abrasion resistance.

**Figure 6 polymers-15-04584-f006:**
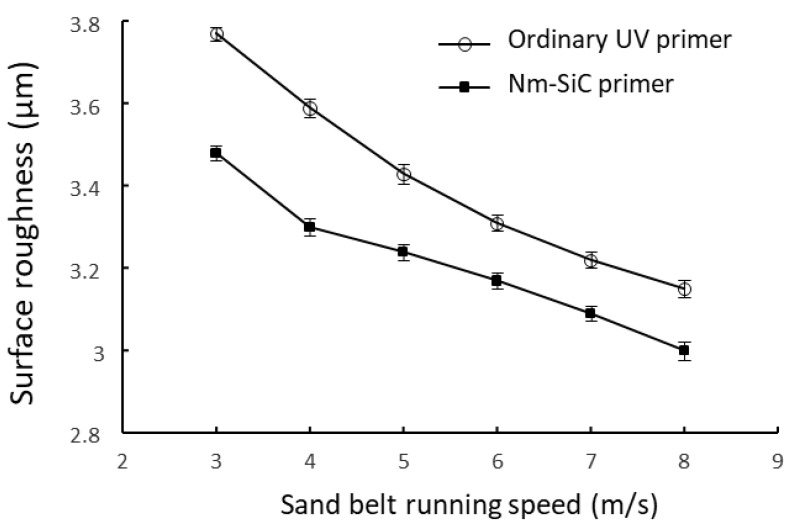
Effect of number of revolutions of sanding belt on surface roughness.

**Figure 7 polymers-15-04584-f007:**
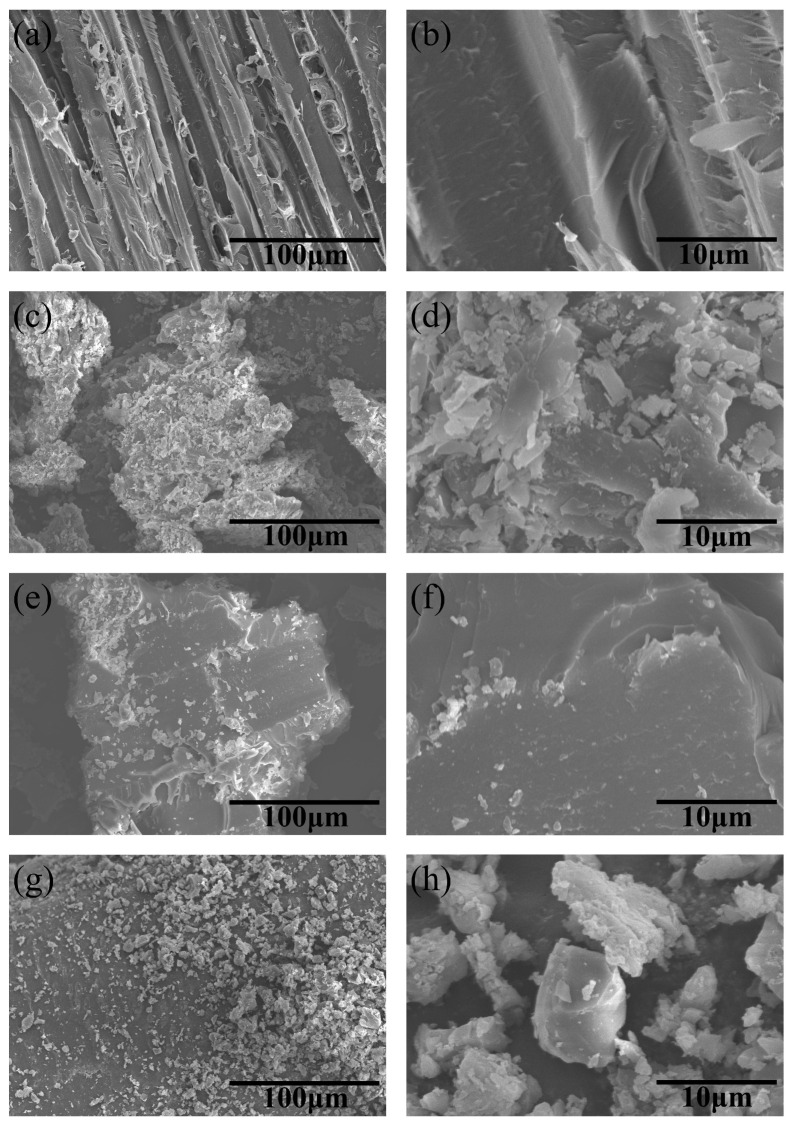
SEM images of wood flooring substrate (**a**,**b**), conventional primer coated flooring (**c**,**d**), conventional paint coated flooring (**e**,**f**), and flooring modified with nano silicon carbide paint (**g**,**h**).

**Figure 8 polymers-15-04584-f008:**
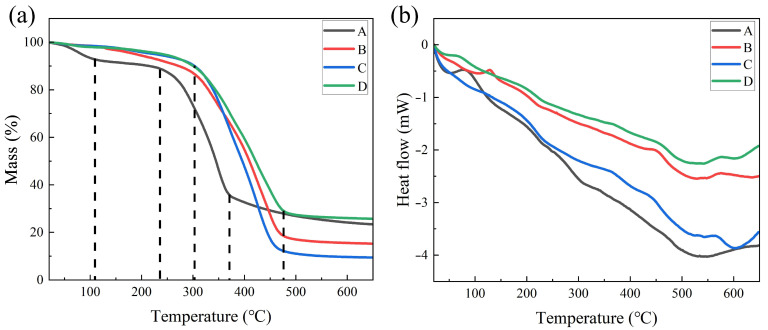
TG curves (**a**) and DSC curves (**b**) of wood flooring substrate (A), conventional primer coated flooring (B), conventional paint coated flooring (C), and flooring modified with nano silicon carbide paint (D).

**Table 1 polymers-15-04584-t001:** Impact of different nano-SiC powder additions on the performance of paint film.

SiC (wt%)	Film Abrasion Resistance (r): (Min, Max)	Abrasion Value (g/100 r): (Min, Max)	Adhesion of Paint Film (Level): (Min, Max)
0	500 (500, 500)	0.134 (0.129, 0.140)	1 (1, 1)
1.0	1800 (1700, 1900)	0.112 (0.103, 0.119)	1 (1, 1)
1.5	2300 (2200, 2400)	0.093 (0.089, 0.100)	1 (1, 1)
2.0	2800 (2700, 2900)	0.072 (0.068, 0.077)	1 (1, 1)
2.5	2400 (2200, 2500)	0.082 (0.077, 0.089)	1 (1, 1)
3.0	1900 (1800, 2100)	0.111 (0.099, 0.117)	1 (1, 1)

**Table 2 polymers-15-04584-t002:** Impacts of different humidity levels on the surface appearance quality of nano-SiC paint film.

RH %	45	55	65	75	85
Paint film’s surface appearance	Flat, smooth, transparent	Flat, smooth, transparent	Flat, smooth, transparent	Flat, smooth, slightly white	White, and occasional slight blistering
Pictures of paint film’s surface appearance	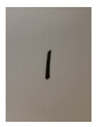	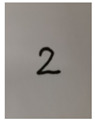	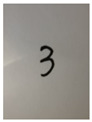	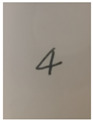	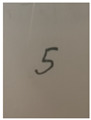

**Table 3 polymers-15-04584-t003:** Impact of coating dosage on the adhesion of paint film and film hardness.

Total Coating Dosage (g/m^2^)	20	30	40	60	80	100	120	160	200	240	280
Adhesion of paint film (level)	1	1	1	2	3	3	4	4	4	4	4
Film hardness	5H	5H	5H	5H	5H	5H	5H	5H	5H	5H	5H

**Table 4 polymers-15-04584-t004:** Performance of SiC paint film on engineered wood flooring.

Test Item	Test Result	Test Method
Quality of surface appearance of paint film	All surfaces are flat and smooth and fully displaying wood texture	Visual inspection
Adhesion of paint film (level) (Min, Max)	0 (0, 1)	GB/T 4893.4-2013
Abrasion value (g/100 r) (Min, Max)	0.072 (0.069, 0.074)	GB/T 17657-2022 4.47
Film abrasion resistance (r) (Min, Max)	2900 (2800, 3000)	GB/T 17657-2022 4.46
Film hardness (Min, Max)	6H (6H, 6H)	GB/T 6739-2022
Surface performance of cycled heat-and-cold resistance	All surfaces are free of cracks, bubbles, discoloring, or shrinkage, etc.	GB/T 17657-2022 4.41 (with the number of cycles revised from two to ten times)

## Data Availability

The data presented in this study are available on request from the corresponding author.

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
