# Peer review of "Preparation of Nano Silicon Carbide Modified UV Paint and Its Application Performance on Wood Flooring Surface"

_polymers, 2023, doi:10.3390/polym15234584_

Round 1

Reviewer 1 Report

Comments and Suggestions for Authors

The main objective of the study was to determine the effect of nano SiC on selected properties of UV varnish product coatings. To achieve this goal, samples with homogeneous dispersion of the nano additive in the polymer matrix were prepared and a test program was developed that included verification of appearance, adhesion, abrasion resistance, variable temp, hardness. In addition, a scanning electron microscopy method was used. The scope of the work was presented in a concise manner. The applied research methods made it possible to demonstrate the influence of the proportion of nanoparticles in the studied varnish coatings. My evaluation of the work is high. The issues of modification with nanoparticles are all the time relevant. The introduction of additives to polymers makes it possible to create new types of composites with unprecedented properties. The authors should make changes and additions to the text of the article.
1. key words: the reviewer suggests to add: "UV lacquer", „nano silicon carbide“, „resistance“.
2. Citation of studies by other authors in the introduction was missing. The state of the art should be supplemented with publications on the problems of UV lacquer products and the obtained coatings.  There is a lack of literature sources of the authors. Readers want to know the achievements of the authors in this problematic (e.g. the results of previous studies).
3. Unsatisfactory is the lack of the characterization of wood in the experimental part. Please provide basic information about the species used in the study (e.g. density, moisture content) and their dimensions.
4. A shortcoming of the work is the lack of indication of the number of samples used in the experiments carried out. This is an important factor taken into account in the statistical estimation of the results of the study and the formulation of conclusions.
5. The methods of research are described in a general way - this part needs additions.
6. The curves in Fig. 2-5 should be described by equations with the correlation coefficient R2.
7. Relationships from thermogravimetric analysis (TG and DSC curves in Fig. 7) should be described with the characteristic temperatures.
8. Information on the number of measurements taken on each sample is missing - please supplement.
9. The authors cited only 17 publications (by scientists from Asia). Is this topic not addressed by other scientists on other continents? Please clarify this and make additions. Double numbering is not necessary (please standardize it).
Based on the results of the study, the authors correctly drew conclusions (however, they were formulated in a general way).

I recommend the article for publication after the above-mentioned additions.

Author Response

Reviewer #1: The main objective of the study was to determine the effect of nano SiC on selected properties of UV varnish product coatings. To achieve this goal, samples with homogeneous dispersion of the nano additive in the polymer matrix were prepared and a test program was developed that included verification of appearance, adhesion, abrasion resistance, variable temp, hardness. In addition, a scanning electron microscopy method was used. The scope of the work was presented in a concise manner. The applied research methods made it possible to demonstrate the influence of the proportion of nanoparticles in the studied varnish coatings. My evaluation of the work is high. The issues of modification with nanoparticles are all the time relevant. The introduction of additives to polymers makes it possible to create new types of composites with unprecedented properties. The authors should make changes and additions to the text of the article.

Response: Thank you for your kindest comments and suggestions. 

  1. key words: the reviewer suggests to add: "UV lacquer", „nano silicon carbide“, „resistance“.

Response: The keywords suggested were added.

Keywords: Super-abrasion-resistant; coating; paint; UV lacque ; wood flooring; nano silicon carbide

  1. Citation of studies by other authors in the introduction was missing. The state of the art should be supplemented with publications on the problems of UV lacquer products and the obtained coatings.There is a lack of literature sources of the authors. Readers want to know the achievements of the authors in this problematic (e.g. the results of previous studies).

Response: Citation of studies by other authors in the introduction was added.More references have also been added, please refer to the introduction section for detail.

  1. Unsatisfactory is the lack of the characterization of wood in the experimental part. Please provide basic information about the species used in the study (e.g. density, moisture content) and their dimensions.

Response:The relevant result information of wood has been supplemented and modified as follows.

The test sample is multi-layer structural composite floor with a specification of 910mm×125mm×15mm, a surface panel of oak (thickness 0.6mm), an intermediate substrate layer of eucalyptus plywood, and a bottom plate of birch veneer.The moisture content of the floor is 9.5%.
4. A shortcoming of the work is the lack of indication of the number of samples used in the experiments carried out. This is an important factor taken into account in the statistical estimation of the results of the study and the formulation of conclusions.

Response:The amount of test samples were shown in the method part. And each level was repeated 3 times. Please refer to the revised manuscript for details.
5. The methods of research are described in a general way - this part needs additions.

Response:The methods of research are described more clearly. Fig.1 and Fig.2 have been redrawn.
6. The curves in Fig. 2-5 should be described by equations with the correlation coefficient R2.

Response:Thank you for your suggestion. Figures 2 to 5 depict the trend of change and do not have relevant equations to fit. Therefore, the correlation coefficient R2 is not given. But thank you still for your suggestion
7. Relationships from thermogravimetric analysis (TG and DSC curves in Fig. 7) should be described with the characteristic temperatures.

Response:The thermogravimetric analysis was more clearly by Fig.8.We have made modifications and descriptions to the images.

  1. Information on the number of measurements taken on each sample is missing - please supplement.

Response: Each level was repeated 3 times. According to your advice, we have provided additional explanations.
9. The authors cited only 17 publications (by scientists from Asia). Is this topic not addressed by other scientists on other continents? Please clarify this and make additions. Double numbering is not necessary (please standardize it).

Response:We have supplemented relevant references from scientists outside of Asia. In addition, relevant standards(GB:Chinese national standard testing methods) will also be supplemented as references. Detailed revisions have been made to the references accordingly.
Based on the results of the study, the authors correctly drew conclusions (however, they were formulated in a general way).

Response:The description of the conclusion has been modified and supplemented.
I recommend the article for publication after the above-mentioned additions.

Response:Thank you very much for your recognition and support of this research work.

Reviewer 2 Report

Comments and Suggestions for Authors

This is an interesting paper with a strong applicative character. However, it needs some improvements. The technical ones (some references are not cited in the paper, the standards should be cited in the same way as other references). Some descriptions should be written more clear (preparation of the coated samples), and experimental part should be amended. See more detailed comments.

Polymers (MDPI): polymers-2633158

Preparation of Nano Silicon Carbide modified UV Paint and Its Application Performance on Wood Flooring Surface

REVIEWER’S REPORT

 Specific comments

2.1 Materials and instruments:

There are some materials mentioned in the paper, that are not mentioned and described in this section. For instance putty. What kind of putty did you use, who was the producer, etc. Also – this was not clear to me – see my next comment, later in the paper, nano-SiC-primer is mentioned for several times. Was this a specially purchased product, or did you prepare it by yourself, and if yes, how. What were the properties of the primer compared to the UV coating (e.g. dry matter, viscosity, etc.)

2.3 Preparation of super…:

See my previous comment – the putty mentioned here is not explained in the part where materials are described. Also, nano-SiC-primer is mentioned for quite some time, but nothing is written in the paper on how this primer was prepared. So, add this description.

Also, the whole preparation is described relatively complicated. One must very carefully read this part to understand it. Please write this part to make it understandable more easy. I suggest that you present all phases in the preparation process in a Table, or maybe in a Scheme, similarly to the one presenting preparation of the SiC UV coating.

2.4 Method for testing:

Quite some test methods are listed and briefly described in this paragraph. But they are not cited and written in the list of references. Please cite all standards in the paper just as all other regular references!

3.3 Impact of humidity:

from what is written in the paper about the influence of humidity, it is not quite clear if the humidity during use (exposure conditions) is meant, or humidity during application and curing. It seems that during application and curing process, but please make this very clear in the paper. And also, I guess you mean the relative air humidity – please make this clear!

3.5 Impact of sanding:

it is not at all clear which sanding do you have in minds. When we look at the specimen preparation description, we can see that the sanding operations were performed for several times. At first there was sanding of the substrate, afterwards of the primer + putty, then of the “sanding primer” (by the way, this term is not understandable, so please clarify it somewhere in the paper), and then another primer-putty cycle was applied, etc. So, please make very clear impact of which stage of sanding did you have in minds?

Figure 5:                       

what does it mean “Nm-SiC primer”? I guess “Nano-SiC primer”. Why yo do not write simply “Nano-SiC primer”?

References:

where in the paper the references [7] - [17] are cited? I cannot find them. Please resolve this problem!

Comments on the Quality of English Language

I think some minor improvements of the quality of English language should be carried out.

Author Response

Reviewer #2: This is an interesting paper with a strong applicative character. However, it needs some improvements. The technical ones (some references are not cited in the paper, the standards should be cited in the same way as other references). Some descriptions should be written more clear (preparation of the coated samples), and experimental part should be amended. See more detailed comments.

Response:Thank you very much for your recognition and support of this research work.

Polymers (MDPI): polymers-2633158

Preparation of Nano Silicon Carbide modified UV Paint and Its Application Performance on Wood Flooring Surface

REVIEWER’S REPORT

 Specific comments

2.1 Materials and instruments:

There are some materials mentioned in the paper, that are not mentioned and described in this section. For instance putty. What kind of putty did you use, who was the producer, etc. Also – this was not clear to me – see my next comment, later in the paper, nano-SiC-primer is mentioned for several times. Was this a specially purchased product, or did you prepare it by yourself, and if yes, how. What were the properties of the primer compared to the UV coating (e.g. dry matter, viscosity, etc.)

Response:According to your suggestions have been modified, see the material section.

The test sample is multi-layer structural composite floor with a specification of 910mm×125mm×15mm, a surface panel of oak (thickness 0.6mm), an intermediate substrate layer of eucalyptus plywood, and a bottom plate of birch veneer.The moisture content of the floor is 9.5%.The UV paint was provided by Hunan Changsha Xinkai Chemical Co., Ltd. The UV paint is mainly composed of polyurethane acrylate, with a content of about 55%. The other components include acrylic monomers (tripropylene glycol diacrylate, trimethylolpropane triacrylate, 1,6-ethylene glycol diacrylate, hydroxyethyl methacrylate,with a content of about 25%), talc powder(about 8%), aromatic ketone initiators (about 5%), polysiloxane defoamers(about 2%), anti precipitation agents(about 3%), leveling agents(about 2%), etc. The viscosity of the UV paint used for coating was 1000-2000mpa·s.The nano-SiC was purchased from Shanghai Chaowei Nanotechnology Co., Ltd. The average particle size of nano silicon carbide is 40 nm, with a purity greater than 99.9% and a specific surface area of 39.8 m2/g. The putty was provided by Sankeshu Coatings Co., Ltd. The viscosity of the putty was 10000-13000 mpa·s. The fully automatic homogenization system was from RayKol (USA). The universal mechanical testing machine was from Shimadzu (Japan). The surface roughness measuring equipment was from Mitutoyo (Japan). The paint film grinding instrument was from Shanghai Rongjida Instrument Technology Co., Ltd.

2.3 Preparation of super…:

See my previous comment – the putty mentioned here is not explained in the part where materials are described. Also, nano-SiC-primer is mentioned for quite some time, but nothing is written in the paper on how this primer was prepared. So, add this description.

Also, the whole preparation is described relatively complicated. One must very carefully read this part to understand it. Please write this part to make it understandable more easy. I suggest that you present all phases in the preparation process in a Table, or maybe in a Scheme, similarly to the one presenting preparation of the SiC UV coating.

Response:The methods of research are described more clearly. Fig.1 and Fig.2 have been redrawn.

2.4 Method for testing:

Quite some test methods are listed and briefly described in this paragraph. But they are not cited and written in the list of references. Please cite all standards in the paper just as all other regular references!

Response:The relevant standards(GB:Chinese national standard testing methods) will also be supplemented as references. Detailed revisions have been made to the references accordingly.

3.3 Impact of humidity:

from what is written in the paper about the influence of humidity, it is not quite clear if the humidity during use (exposure conditions) is meant, or humidity during application and curing. It seems that during application and curing process, but please make this very clear in the paper. And also, I guess you mean the relative air humidity – please make this clear!

Response:The humidity was the relative air humidit during application and curing. We have made modifications and explanations, please refer to the revised manuscript for details

3.5 Impact of sanding:

it is not at all clear which sanding do you have in minds. When we look at the specimen preparation description, we can see that the sanding operations were performed for several times. At first there was sanding of the substrate, afterwards of the primer + putty, then of the “sanding primer” (by the way, this term is not understandable, so please clarify it somewhere in the paper), and then another primer-putty cycle was applied, etc. So, please make very clear impact of which stage of sanding did you have in minds?

Response:The methods of research are described more clearly in Fig.2. Sand belt 400 # (Represented by the step 11 in the Fig. 2) was selected to compare the impact of the sanding belt's operating speed on the surface roughness of the floorings' nano-SiC primer.

Figure 5:what does it mean “Nm-SiC primer”? I guess “Nano-SiC primer”. Why yo do not write simply “Nano-SiC primer”?

Response: We have modified it as you suggested.

References:where in the paper the references [7] - [17] are cited? I cannot find them. Please resolve this problem!

 Response:We have supplemented relevant references from scientists outside of Asia. In addition, relevant standards(GB:Chinese national standard testing methods) will also be supplemented as references. Detailed revisions have been made to the references accordingly.

Comments on the Quality of English Language

I think some minor improvements of the quality of English language should be carried out.

 Response:The English of the full text has been modified and improved. Thank you for your suggestions.

Round 2

Reviewer 2 Report

Comments and Suggestions for Authors

The paper is OK now, I am satisfied with the answers to my review in the first review round. Only the Quality of English written language should be improved - there are quite some typing errors. See for instance Fig. 2, where "Paint" is written for several times wrongly as "Piant". There are some other typing errors elsewhere in the paper - for example, in the key words it is written "UV lacque" instead of "UV lacquer" as it should be written. Etc.

Comments on the Quality of English Language

The Quality of English written language should be improved - there are quite some typing errors. See for instance Fig. 2, where "Paint" is written for several times wrongly as "Piant". There are some other typing errors elsewhere in the paper - for example, in the key words it is written "UV lacque" instead of "UV lacquer" as it should be written. Etc.

Author Response

Thank you very much for your comments and valuable feedback on this research work.We have consulted experts from native language countries to revise the English version of the entire text. Especially in response to some spelling errors you raised, we have made revisions one by one. Thank you again.
